# Economic Evaluation of Nemonoxacin, Moxifloxacin and Levofloxacin in the Treatment of Early Community-Acquired Pneumonia with Possible Pulmonary Tuberculosis

**DOI:** 10.3390/ijerph19084816

**Published:** 2022-04-15

**Authors:** Mingye Zhao, Zhuoyuan Chi, Xingming Pan, Yue Yin, Wenxi Tang

**Affiliations:** 1School of International Pharmaceutical Business, China Pharmaceutical University, Nanjing 211198, China; 3220040596@stu.cpu.edu.cn (M.Z.); 3220040603@stu.cpu.edu.cn (Z.C.); 3320041623@stu.cpu.edu.cn (X.P.); 3321041182@stu.cpu.edu.cn (Y.Y.); 2Center for Pharmacoeconomics and Outcomes Research, China Pharmaceutical University, Nanjing 211198, China

**Keywords:** community-acquired pneumonia, cost-utility analysis, pulmonary tuberculosis, levofloxacin, nemonoxacin, moxifloxacin

## Abstract

The Chinese community-acquired pneumonia (CAP) Diagnosis and Treatment Guideline 2020 recommends quinolone antibiotics as the initial empirical treatment options for CAP. However, patients with pulmonary tuberculosis (PTB) are often misdiagnosed with CAP because of the similarity of symptoms. Moxifloxacin and levofloxacin have inhibitory effects on mycobacterium tuberculosis as compared with nemonoxacin, resulting in delayed diagnosis of PTB. Hence, the aim of this study is to compare the cost-effectiveness of nemonoxacin, moxifloxacin and levofloxacin in the treatment of CAP and to determine the value of these treatments in the differential diagnosis of PTB. Primary efficacy data were collected from phase II-III randomized, double-blind, multi-center clinical trials comparing nemonoxacin to moxifloxacin (CTR20130195) and nemonoxacin to levofloxacin (CTR20140439) for the treatment of Chinese CAP patients. A decision tree was constructed to compare the cost-utility among three groups under the perspective of healthcare system. The threshold for willingness to pay (WTP) is 1–3 times GDP per capita ($11,174–33,521). Scenarios including efficacy and cost for CAP patients with a total of 6% undifferentiated PTB. Sensitivity and scenario analyses were performed to test the robustness of basic analysis. The costs of nemonoxacin, moxifloxacin, and levofloxacin were $903.72, $1053.59, and $1212.06 and the outcomes were 188.7, 188.8, and 188.5 quality-adjusted life days (QALD), respectively. Nemonoxacin and moxifloxacin were dominant compared with levofloxacin, and the ICER of moxifloxacin compared with nemonoxacin was $551,643, which was much greater than WTP; therefore, nemonoxacin was the most cost-effective option. Regarding patients with PTB who were misdiagnosed with CAP, taking nemonoxacin could save $290.76 and $205.51 when compared with moxifloxacin and levofloxacin and resulted in a gain of 2.83 QALDs. Our findings demonstrate that nemonoxacin is the more economical compared with moxifloxacin and levofloxacin, and non-fluoroquinolone antibiotics are cost-saving and utility-increasing compared to fluoroquinolones in the differential diagnosis of PTB, which can help healthcare system in making optimal policies and help clinicians in the medication of patients.

## 1. Introduction

Community-acquired pneumonia (CAP) refers to inflammation or infection of the lung parenchyma that is contracted outside the hospital, including pneumonia that has a clear incubation period and pathogen infection that develops during the incubation period after admission [1]. CAP is a common disease of the respiratory system; it has a high incidence worldwide and brings a considerable economic burden to society [2,3,4,5,6]. The incidence of CAP in China is higher than that in European and American countries; owing to the combination of urban air pollution and low vaccination rates of residents [7,8], China is still facing the challenges of preventing and treating CAP and the associated burdens of the disease.

The clinical manifestations of CAP and pulmonary tuberculosis (PTB) are similar, and the imaging examination results of the two diseases are diverse. The positive rate for pathogen culture detection is low and the process is time-consuming, which makes it difficult to distinguish between the two diseases [9,10]. Among the patients with CAP in China, approximately 6% of this population comprises patients with PTB, as shown by expert investigation, which is higher than the Asian average of 3.30% [11]. The main treatment for CAP is antibiotic therapy. Levofloxacin and moxifloxacin are typical fluoroquinolone drugs with strong, broad-spectrum antibacterial effects [7,8]. However, studies have found that the use of fluoroquinolone antibiotics in the diagnosis and treatment of patients with CAP with unclear symptoms increases the risk of delaying the diagnosis of PTB [12]. Hence, some experts do not recommend the use of fluoroquinolone antibiotics as first-line treatment for CAP [13]. Nemonoxacin is a new type of non-fluoroquinolone drug that maintains strong antibacterial activity against common CAP-causing pathogens, such as Streptococcus pneumoniae, Staphylococcus aureus, Haemophilus influenzae, and Moraxella catarrhalis [14]. In addition to its clinically satisfactory cure rate and bacterial clearance rate, nemonoxacin has almost no antibacterial effect on Mycobacterium tuberculosis, so it can be used as an early treatment plan for patients with suspected PTB or CAP.

To our knowledge, there are currently no studies in the literature discussing the value of antibiotics in the differential diagnosis of CAP and PTB; furthermore, there is a lack of evidence of the cost-effectiveness of nemonoxacin for the treatment of CAP. Hence, the aim of this study is to compare the cost-effectiveness of nemonoxacin, moxifloxacin, and levofloxacin in the treatment of mild to moderate CAP and to determine the value of these treatments in the differential diagnosis of PTB, and to provide evidence for Chinese governments, clinicians and other researchers in policy making and clinical medication.

## 2. Materials and Methods

### 2.1. Source of Parameters

Primary efficacy data were collected from phase II-III randomized, double-blind, multi-center clinical trials comparing nemonoxacin to moxifloxacin (CTR20130195) and nemonoxacin to levofloxacin (CTR20140439) for treatment of Chinese CAP patients. More clinical trial information is provided in the Appendix A.

For parameters that were unavailable from the literature, we conducted physician surveys. In order to make our data reliable and able to represent the overall level of China, we conducted surveys among authoritative hospitals in the eastern, middle, and western regions of China and collected information by distributing electronic questionnaires to doctors with extensive clinical experience. Finally, 31 doctors from nationwide authoritative hospitals, which were located in Beijing, Shanghai, Wuhan, and other cities, were questioned. The mean years of the clinical treatment experience of these doctors was 20 ± 8 years and their average age was 45 ± 8 years old.

### 2.2. Model Structure

Microsoft Excel 2019 was used to build a decision tree model (Figure 1). The research was conducted from the perspective of the Chinese healthcare system with only direct medical costs considered. The model simulated the end of treatment for 99.9% of patients. The study period was 194 days. The model structure was determined in accordance with disease-specific and expert advice.

The study population consisted of patients with mild and moderate CAP requiring treatment, which included an estimated 6% of patients with PTB who were misdiagnosed with CAP; all patients received one of the three groups of CAP drugs. After the initial CAP injections treatment, those who responded were to receive sequential outpatient or inpatient treatment; during sequential treatment, patients would receive oral therapies (more details are provided in the Appendix A). Nemonoxacin has no inhibitory effect on tuberculosis, whereas levofloxacin and moxifloxacin inhibit M. tuberculosis and mask the symptoms of PTB [12]. Therefore, after the initial treatment of patients with PTB who were misdiagnosed with CAP, patients treated with nemonoxacin were to receive the standard hospitalization treatment for PTB, and the patients in the levofloxacin and moxifloxacin group received sequential CAP treatment. Although levofloxacin and moxifloxacin have been shown to have good anti-tuberculosis effects [15,16], intensive treatment is needed in the early stage of PTB. Therefore, after the short-term CAP-related treatment, patients with PTB would again show symptoms [12]. Considering that the effect may be delayed, patients who did not respond to the initial treatment received second-line treatment or to continue to take the original drug treatment. In the second-line treatment, depending on whether the patient’s condition had deteriorated, it was determined whether the patients should be admitted to general wards or ICU wards. After ICU treatment improved, patients were transferred to the general ward for second-line treatment. Patients who continued the original drug treatment entered the second-line treatment if the course of treatment was not effective. In addition, given the likelihood of the recurrence of CAP [17,18], patients who were cured initially may have been admitted to the hospital again for treatment.

### 2.3. Model Assumptions

To construct the model, the following necessary assumptions were made:That the success of treatment had occurred by the last day of the state.That the recurrence of CAP would have occurred by the 7th day after the initial treatment was effective.That the spread of tuberculosis and delayed treatment (9 days) were not considered to affect the treatment of patients with PTB.That the effect of minor adverse reactions on the utility value and the adverse reactions of the second-line treatment were not considered.That the effective rate of the second-line treatment of patients with recurrent CAP and the effective rate of PTB treatment was 100%.That the proportion of continued injection therapy after the initial treatment failure was the same as the rate of delay in the initial treatment failure.That relapsed patients with CAP were admitted to the hospital for second-line treatment.That patients with PTB relapsed after the CAP sequential treatment ended.

### 2.4. Transition Probability

The initial curative effects of the three groups of drugs were in keeping with the efficacy of Visit 2 (3–5 days of treatment). The data for nemonoxacin, levofloxacin, and moxifloxacin were all derived from phase II–III clinical trials. For patients whose initial treatment failed, 91% and 86% of patients in the moxifloxacin and levofloxacin groups, respectively, chose to continue treatment with the original drug owing to delayed efficacy, which were extracted from similar cost-effectiveness analysis [19]. The delayed rate of efficacy in the nemonoxacin group was determined to be 93% through expert consultation. The recurrence rate after successful initial treatment was calculated from the following equation:(1)$$Recurrence\;rate=\left[\left(1-e_{3}\right)-\left(1-e_{2}\right)\;*\;R\right]/e_{2}.\;$$

In Equation (1), $e_{2}$ and $e_{3}$ represent the effective rates of initial treatment and complete treatment (1–2 days after the end of treatment), respectively, and R represents the rate of delayed efficacy in the population with initial treatment failure.

More details about clinical efficacy, transition probability, treatment time are provided in Table 1.

### 2.5. Cost

From the perspective of the healthcare system, our research only focused on the direct medical costs, which mainly comprised the costs of antibiotics, examinations, auxiliary medication, bed days, nursing, and adverse reaction (AE) treatment. In terms of drug costs, the prices of nemonoxacin, moxifloxacin, and levofloxacin oral and injection solutions and other drugs were determined regarding the national median bid price in 2021. AE was considered to include gastrointestinal disorders (nausea, diarrhea, vomiting, and abdominal discomfort), skin and subcutaneous tissue disorders (rash), neurological disorders (dizziness, headache), and abnormal detection (increased transaminases, decreased WBC, blood bilirubin increase); related treatment costs were from expert consultation. In addition, the second-line treatment drugs were determined through expert consultation, and the dosage of each drug was determined in accordance with the prescriber’s instructions. The value and source of each cost parameter are presented in Table 1; dosing regimens are presented in the Appendix A Appendix A. The costs in this research are in keeping with 2021 values of CNY (1 CNY = 0.15699 USD).

### 2.6. Utility

The range of utility was 0–1; the utility for the death state was 0 and that for the healthy population was 1. Utilities for initial injection therapy in patients with CAP were 0.56, 0.88, and 0.82 for CAP outpatients or inpatients with sequential treatment; for CAP patients in second-line or ICU general wards were 0.53 and 0.30, respectively. For PTB patients, utilities of no treatment, hospital discharge, and hospitalization were 0.68, 0.83, and 0.59, respectively. All utilities were from the published literature. More details are shown in Table 1.

### 2.7. Cost-Effectiveness Analysis

A cost-utility analysis (CUA) was adopted in our study. The effect index used during this study was the quality-adjusted life day (QALD), and the incremental cost-effectiveness ratio (ICER) and the incremental net monetary benefit (INMB) were used to compare the cost-effectiveness of each plan; the calculation method of INMB is shown in Equation (2). According to WHO recommendations, willingness to pay (WTP) was set at 1–3 times GDP per capita (CNY 72,447–217,341). When the ICER was below CNY 217,341, there was a certain cost-effective advantage [32]. Furthermore, we used 1-time GDP as WTP when calculating INMB, and INMB over 0 means economical. Owing to the short research period, we did not consider discounts.
(2)Incremental NMB=WTP∗ΔU−ΔC

### 2.8. Sensitivity Analysis

Taking the uncertainty of the value of each parameter into account, sensitivity analysis was conducted. One-way sensitivity analysis was conducted to explore the economics of each program when the parameters were changed between the upper and lower limits; INMB was used as the economic measure, and the 10 parameters that had the greatest impact on INMB were selected to be plotted as a cyclone graph. Probability sensitivity analysis was conducted using Monte Carlo simulation with 10,000 iterations, a scatter diagram was drawn, and the cost-effective acceptability chart (CEAC) curve was used to analyze the economic situation under different values of WTP.

### 2.9. Scenario Analysis

Similar economic evaluation research tended to ignore the proportion of patients with CAP who are misdiagnosed with PTB [18,21,23,30,33,34]; therefore, to explore the reliability of the conclusions, and to further verify the value of the differential diagnosis of PTB, a scenario analysis in which PTB was not considered was performed.

## 3. Results

### 3.1. Basic Analysis Results

The basic analysis results of the three interventions are provided in Table 2. The cumulative costs of nemonoxacin, moxifloxacin, and levofloxacin were CNY 5859 (USD 904), CNY 6831 (USD 1054), and CNY 7858 (USD 1212), and the cumulative utilities were 188.7, 188.8, and 188.5 QALDs, respectively. In terms of the effectiveness, the cumulative utilities of the three drugs were almost the same. In terms of cost-effectiveness, nemonoxacin and moxifloxacin were dominant compared with levofloxacin, and the ICER of moxifloxacin, compared with nemonoxacin, was CNY 3576,742 (USD 551,643), which was much greater than WTP; therefore, nemonoxacin was the most cost-effective option.

In the value of differential diagnosis of PTB, PTB patients who were misdiagnosed as CAP treated with nemonoxacin, moxifloxacin and levofloxacin would cost CNY 9375 (USD 1472), CNY 11,227 (USD 1763) and CNY 10,684 (USD 1677), respectively. Compared with moxifloxacin and levofloxacin, nemonoxacin saved CNY 1852 (USD 291) and CNY 1309 (USD 205), accounting for approximately 32.7% and 23.1% of the treatment cost of PTB (CNY 5658) [24]. Moreover, treatment with nemonoxacin resulted in an increase of 2.83 QALDs compared to the other two drugs. Further details are presented in Table 2. Furthermore, we decomposed the cumulative costs of these three drugs and the corresponding cost-breakdown table is presented in Table 3.

### 3.2. Results of the Sensitivity Analysis

#### 3.2.1. One-Way Sensitivity Analysis

As shown in Figure 2, the recurrence rate, price of injections, and the number of initial treatment days were the most consequential factors affecting the economy of each drug. From the Figure 2B,C, we can see that nemonoxacin had more NMB than the other drugs when the upper and lower limits of each variable were changed; therefore, nemonoxacin can be considered a cost-effective option compared with moxifloxacin or levofloxacin. Moxifloxacin is more cost-effective compared with moxifloxacin or levofloxacin, as Figure 2A shows.

#### 3.2.2. Probabilistic Sensitivity Analysis

In the scatter plot (Figure 3A), the horizontal axis represents the incremental utilities of nemonoxacin compared with moxifloxacin and levofloxacin and the vertical axis represents the corresponding incremental cost. The dotted lines indicate the values of the two WTPs, respectively, one time and three times per capita GDP. As can be seen from the figure, for moxifloxacin and levofloxacin, 5.28% and 0.03% of all 10,000 points, respectively, at the lower right of the WTP of the three times GDP. That is, compared with moxifloxacin and levofloxacin, nemonoxacin is very cost-effective.

The CEAC (Figure 3B) shows the possibility of being the most cost-effective for each option under different WTPs. It can be seen from the figure that when the WTP fluctuated between CNY 0–300,000, nemonoxacin was always the most economical option. The probabilistic sensitivity analysis results are highly consistent with the basic analysis results, confirming that the basic conclusions are highly reliable; that is, compared with moxifloxacin and levofloxacin, nemonoxacin is the most cost-effective option.

#### 3.2.3. Scenario Analysis

Assuming that the proportion of misdiagnosed PTB in CAP patients is 0, the effectiveness and cost-effectiveness of the results were similar to the basic analysis; that is, nemonoxacin was the most cost-effective solution. Further details are presented in Table 2.

## 4. Discussion

### 4.1. Literature Review

To our knowledge, there are few articles at present that have focused on the economic evaluation of nemonoxacin in the treatment of CAP and there is no literature discussing the value of antibiotics in the differential diagnosis of CAP and PTB. Xiwen [23] discussed the economics of moxifloxacin and levofloxacin in the initial treatment of mild to moderate CAP in accordance with the perspective of Chinese healthcare system. The clinical efficacy of each drug was obtained from a meta-analysis of previous studies, and a CUA model was constructed as per these findings. They found that moxifloxacin showed better effectiveness and cost-effectiveness when compared with levofloxacin. However, the meta-analysis of the literature used in the study is of low quality, and there were large biases in the blinding and randomization. Lloyd [33] used the MOTIV trial that compared the cost-effectiveness of moxifloxacin and levofloxacin combined with ceftriaxone in the treatment of CAP in the German population using cost minimum analysis from the payer perspective, which could not fully measure the comprehensive value of each drug. Cornelis [35] adopted a crossover trial and used a combination of cost minimum analysis and CUA to explore the economic differences between lactam monotherapy, lactam and macrolide combined therapy, and quinolone monotherapy. Quinolone antibiotics were found to be similar to the other two types with no remarkable differences in cost-effectiveness.

### 4.2. The Value of Nemonoxacin in CAP and in the Differential Diagnosis of PTB

The value of nemonoxacin in the differential diagnosis of CAP and PTB is reflected in the fact that nemonoxacin is not effective against *M. tuberculosis* and can therefore isolate potential patients with PTB from people with suspected CAP in the initial treatment stage. Consequently, patients with PTB can be diagnosed and treated in a timely manner, which prevents the delayed diagnosis arising from the masking effect of quinolone drugs on *M. tuberculosis* and thereby avoids the disease deterioration, utility and cost losses caused by the delayed diagnosis. In addition, the trial results showed that the recurrence rate after nemonoxacin treatment was lower than levofloxacin and moxifloxacin, thus the overall effectiveness for the treatment of CAP is more reliable.

Basic analysis results showed that there is little difference in the effectiveness of the three drugs. Compared with moxifloxacin and levofloxacin, treatment with nemonoxacin resulted in savings of CNY 972 (USD 153) and CNY 1999 (USD 314), respectively. The results of the scenario analysis showed that when misdiagnosed PTB was not considered, the economic advantage of nemonoxacin over the other two groups was still obvious, which verified that nemonoxacin was a cost-effective option compared with levofloxacin and moxifloxacin in the treatment of CAP.

PTB patients misdiagnosed with CAP treated with nemonoxacin, moxifloxacin and levofloxacin were not only cost-saving, but also utility-increasing. For value-in-money, nemonoxacin saved CNY 1852 (USD 291) and CNY 1309 (USD 206) compared with moxifloxacin and levofloxacin, respectively, accounting for approximately 32.7% and 23.1% of the total cost of PTB treatment. Simultaneously, for patients with PTB who were misdiagnosed with CAP, the administration of nemonoxacin, compared with moxifloxacin and levofloxacin, caused an increase of 2.83 QALDs. These findings indicated that non-fluoroquinolone antibiotics were of remarkable value in the differential diagnosis of PTB compared with other fluoroquinolone antibiotics.

### 4.3. Strengths and Limitations

First, the study found that the use of fluoroquinolone antibiotics in the diagnosis and treatment of CAP patients with unclear symptoms increases the risk of delaying the diagnosis of tuberculosis [12]. Second, in the previous articles studying the economics of CAP drugs, the misdiagnosis of tuberculosis patients had not been considered [18,21,23,31,33,34]. This study considered that a certain proportion of patients with CAP were misdiagnosed with tuberculosis and performed a CUA analysis on this basis. In addition, the clinical path of our research is more comprehensive, and the research results are more in line with the “real world” situation. For example, previous studies have ignored recurrence in patients with successful initial treatment [23]; however, the clinical trial results showed that the effective rate of each drug at Visit 3 was substantially lower than that at Visit 2. Consequently, we believe that in addition to the factors of the change of trial personnel, patients for whom the drug is effective at the initial stage of treatment are likely to relapse in the later stage of treatment or after the end of treatment [34]. Finally, the clinical efficacy data of this study were derived from a multicenter, randomized, double-blind, parallel-controlled randomized controlled trial, which has high reliability and is very suitable for CAP-related evaluations based on the Chinese population.

This study also has certain limitations. First, the relevant costs, such as hospitalization fees, examination fees, and nursing fees, lack a national representative data set; the costs are instead derived from the survey results of experts from different regions with a limited sample size and may not be representative of the whole country. Therefore, the relevant parameters in the model are slightly biased. However, considering that this bias is the same among all groups, it should not affect the analysis results. Second, owing to the lack of utility data for hospitalized patients with CAP, this study adopted the utility value used in hospital-acquired pneumonia-related studies. Considering that the antibiotic treatment plan used and the patients’ conditions are similar [23], we believe that the utilities of these studies are the best source of data. It should be noted that the abovementioned factors in this study have been subjected to sensitivity analyses and the results of these analyses showed that these assumptions should not affect the research conclusions. Third, the indirect costs ignored due to our choice of the healthcare system perspective and inaccessibility of relevant data, may lead to bias in cost estimates. However, due to the short duration of the study, the indirect costs such as loss of work, transportation, and nutrition were roughly the same across groups, so it would not have had an impact on our conclusion.

## 5. Conclusions

Our study reveals that nemonoxacin is more economical than moxifloxacin and levofloxacin in the treatment of mild to moderate CAP, and we also confirmed the values in money and utility of non-fluoroquinolone antibiotics in the differential diagnosis of pulmonary tuberculosis compared with fluoroquinolone antibiotics. Due to the reduced treatment cost, compared with moxifloxacin and levofloxacin, nemonoxacin is the most cost-effective option in the initial treatment of CAP under the current WTP. For PTB patients misdiagnosed as CAP, treatment with nemonoxacin caused an average saving of CNY 1600 (USD 251.2) compared with fluoroquinolone antibiotics, about 28.3% of total treatment cost of PTB, as well as an increase of 2.83 QALDs in utility. These results may help healthcare system make relevant policies and help clinicians in choosing optimal drugs in treating CAP. Further research based on patient or society perspective with a more precise calculation of both direct and indirect cost is needed to provide additional information about more non-fluoroquinolone and fluoroquinolone antibiotics in the treatment of CAP or misdiagnosed PTB.

## Figures and Tables

**Figure 1 ijerph-19-04816-f001:**
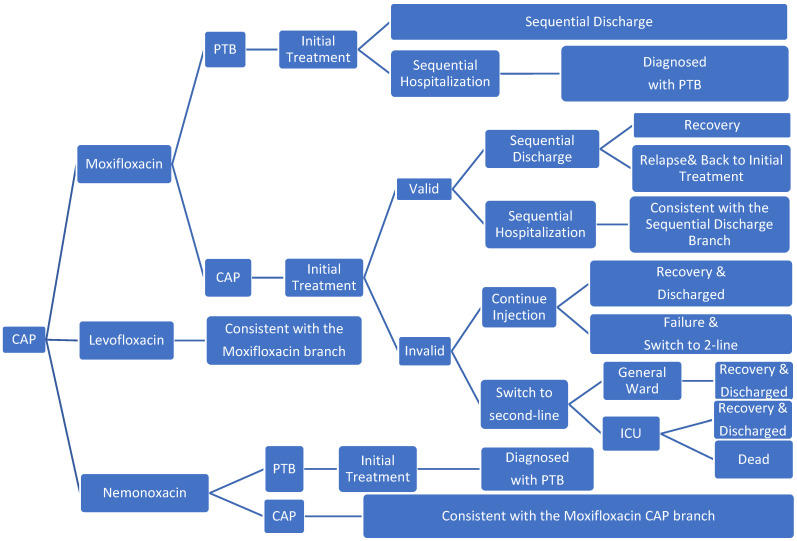
Model structure (PTB, pulmonary tuberculosis).

**Figure 2 ijerph-19-04816-f002:**
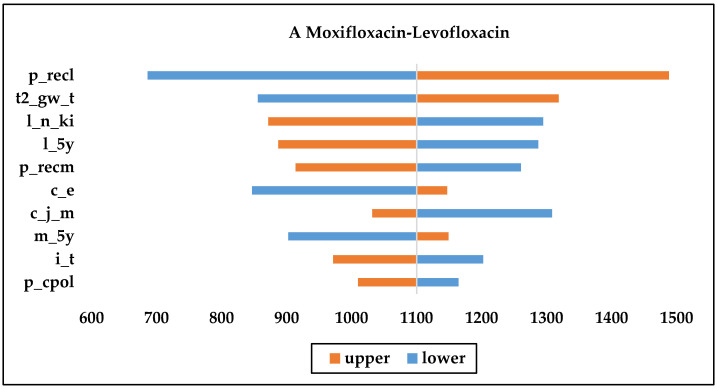
One-way sensitivity analysis chart. (c_j_m: daily cost of moxifloxacin injection; c_j_n: daily cost of nemonoxacin injection; c_e: daily cost of second-line therapy; m_5y: Moxifloxacin initial treatment effective rate; l_5y: Levofloxacin initial treatment effective rate; n_5y: Nemonoxacin initial treatment effective rates; n_ki: proportion of original drug treatment after initial failure of levofloxacin; n_n_ki: proportion of original drug treatment after initial failure of nemonoxacin; i_t: initial treatment day; t2_gw_t: second-line treatment days in general; p_recm: recurrence rate for moxifloxacin after effective initial treatment; p_recl: recurrence rate for levofloxacin after effective initial treatment; p_recn: recurrence rate for nemonoxacin after effective initial treatment; p_cpol: initial treatment effect delay rate for levofloxacin; p_cpon; initial treatment effect delay rate for nemonoxacin; m_y_ko: sequential proportion of hospitalizations after successful initial treatment with moxifloxacin).

**Figure 3 ijerph-19-04816-f003:**
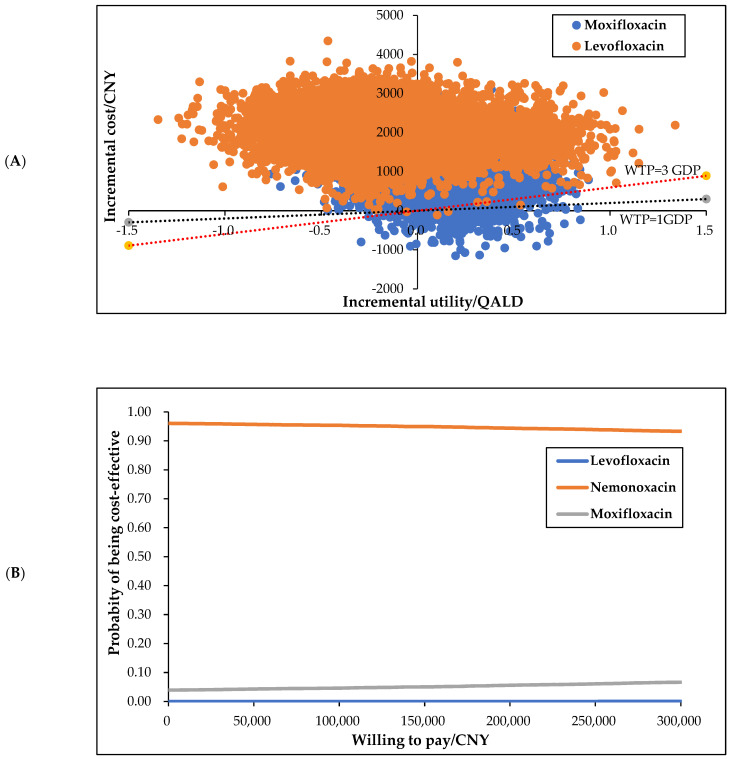
Scatter plot (**A**) and cost-effective acceptability plot (**B**).

**Table 1 ijerph-19-04816-t001:** Input parameters used in the analysis.

Parameters	Mean	Range	Disturbution	Source
Clinical inputs				
Initial treatment effective rate (moxifloxacin)	100%	96–100%	beta	Phase II-III RCT
Initial treatment effective rate (levofloxacin)	93%	90–96%	beta	
Initial treatment effective rate (nemonoxacin)	94%	92–97%	beta	
Complete treatment efficiency (moxifloxacin)	97%	——	fixed	
Complete treatment efficiency (levofloxacin)	87%	——	fixed	
Complete treatment efficiency (nemonoxacin)	93%	——	fixed	
Sequential proportion of hospitalizations after successful initial treatment (moxifloxacin)	87%	83–91%	beta	[20], Physician survey
Sequential proportion of hospitalizations after successful initial treatment (levofloxacin)	97%	92–98%	beta	
Sequential proportion of hospitalizations after successful initial treatment (nemonoxacin)	77%	76–79%	beta	
Proportion of using original treatment after initial failure (moxifloxacin)	91%	86–96%	beta	[19], Physician survey
Proportion of using original treatment after initial failure (levofloxacin)	86%	82–90%	beta	
Proportion of using original treatment after initial failure (nemonoxacin)	93%	89–98%	beta	
Recurrence rate after effective initial treatment (moxifloxacin)	3%	3–4%	beta	Calculate
Recurrence rate after effective initial treatment (levofloxacin)	8%	6–10%	beta	
Recurrence rate after effective initial treatment (nemonoxacin)	2%	1–4%	beta	
Initial treatment effect delay rate (moxifloxacin)	91%	86–96%	beta	[18], Physician survey
Initial treatment effect delay rate (levofloxacin)	86%	82–90%	beta	
Initial treatment effect delay rate (nemonoxacin)	93%	89–98%	beta	
Proportion of general wards in second-line treatment	91%	86–96%	beta	[21]
ICU treatment mortality	10%	10–11%	beta	[22]
Proportion of PTB in CAP	6%	5–7%	beta	Physician survey
Cost/CNY				
Injection (moxifloxacin)	127.22	175.33–230.12	gamma	Local market ^b^
Injection (levofloxacin)	68.83	83.21–109.21	gamma	
Injection (nemonoxacin)	84.8	67.84–89.04	gamma	
Sequential drug (moxifloxacin)	3.94	3.15–4.14	gamma	Local market ^b^
Sequential drug (levofloxacin)	2.6	2.08–2.73	gamma	
Sequential drug (nemonoxacin)	32.4	25.92–34.02	gamma	
Initial treatment days	5.0	4.0–6.0	gamma	Physician survey
Sequential treatment days	6.0	4.8–7.2	gamma	
Daily cost of second-line therapy	2233.76	1839.6–2414.48	gamma	Local market ^b^, [23]
Daily cost for second-line treatment inspection: Bronchoscopy	1761	1672.95–1849.05	gamma	[23]
Daily cost for second-line treatment inspection: NGS ^a^	3541.01	3363.95–3718.05	gamma	
Additional cost during ICU	4492.95	4268.31–4717.60	gamma	
Hospital inspection fee for first-line treatment	2205.13	2094.75–2315.25	gamma	Physician survey
Daily cost of inpatient supplementary medication	110.25	88.20–115.76	gamma	
Daily cost of outpatient supplementary medication	30.87	29.33–32.41	gamma	
Total cost of inpatient bed	55.13	52.37–57.89	gamma	
Total cost of hospital care	33.08	31.49–34.73	gamma	
Total treatment cost of PTB	5658.08	5375.18–5940.98	gamma	[24]
Treatment cost of nemonoxacin injection AE ^c^	95.36	90.59–100.13	gamma	Physician survey
Treatment cost of moxifloxacin injection AE ^c^	145.67	138.39–152.95	gamma	
Treatment cost of levofloxacin injection AE ^c^	67.45	64.08–70.82	gamma	
Treatment cost of levofloxacin capsules AE ^c^	64.83	61.59–68.07	gamma	
Treatment cost of moxifloxacin capsules AE ^c^	45.99	43.69–48.29	gamma	
Treatment cost of nemonoxacin capsules AE ^c^	84.38	80.16–88.60	gamma	
Days of treatment using the original drug when initial treatment failed	6	4.8–9.0	gamma	
Second-line treatment days in general ward	9.0	7.2–10.8	gamma	[21]
Days of second-line treatment in ICU ward	8.0	6.4–9.6	gamma	[20]
Days of treatment after ICU transferred to general ward	13.0	10.4–15.6	gamma	[23], Physician survey
Days of hospitalization for PTB	9.5	7.0–11.0	gamma	[25,26,27]
Intensive treatment days for PTB outpatient	50.5	49.0–53.0	gamma	[25,26,27]
Consolidation treatment days for PTB outpatient	120.0	114.0–126.0	gamma	[27]
Utility				
Utility for initial injection therapy in patients with CAP	0.56 (0.51–0.62)	0.51–0.62	beta	[28]
Utility for CAP outpatients with sequential treatment	0.88 (0.79–0.97)	0.79–0.97	beta	[29]
Utility for CAP inpatients with sequential treatment	0.82 (0.74–0.90)	0.74–0.90	beta	[30]
Utility for CAP patients in second-line general ward	0.53 (0.48–0.58)	0.48–0.58	beta	[30]
Utility for CAP patients in ICU general ward	0.3 (0.27–0.33)	0.27–0.33	beta	[30]
Utility for PTB outpatients	0.83 (0.50–0.98)	0.50–0.98	beta	[31]
Utility for PTB inpatients	0.59 (0.47–0.71)	0.47–0.71	beta	[31]
Utility for PTB patients without treatment	0.68 (0.65–0.72)	0.65–0.72	beta	[28]

^a^ NGS, next-generation sequencing; ^b^ https://www.yaozh.com/ accessed on 11 November 2021; ^c^ AE, adverse reactions.

**Table 2 ijerph-19-04816-t002:** Basic analysis and scenario analysis results.

Basic Analysis						
Treatment	Cost (CNY)	Cost for PTB Patients Misdiagnosed as CAP (CNY)	Utility (QALD)	Utility for PTB Patients Misdiagnosed as CAP (QALD)	ICER I	ICER II
Nemonoxacin	5859.55 (USD 903.72)	9374.67 ($1471.83)	188.71	159.00	——	——
Moxifloxacin	6831.26 (USD 1053.59)	11,226.50 ($1762.56)	188.80	156.17	3546,742 (USD 551,642)	——
Levofloxacin	7858.78 (USD 1212.06)	10,684.00 ($1677.39)	188.51	156.17	dominated	dominated
Scenario analysis (assuming proportion of misdiagnosed PTB in CAP patients is 0)						
Nemonoxacin	5635.18 (USD 884.72)	——	190.60	——	——	——
Moxifloxacin	6550.71 (USD 1028.46)	——	190.69	——	3940,823 (USD 607,795)	——
Levofloxacin	7678.45 (USD 1205.52)	——	190.37	——	dominated	dominated

**Table 3 ijerph-19-04816-t003:** Breakdown for basic analysis costs (CNY).

Group	First-Line Injections and Sequential Drugs	Hospitalization for First- and Second-Line	Adverse Events	Second-Line Drugs	ICU Treatment	Tuberculosis Treatment	Total Cost
Nemonoxacin	508.22 (USD 79.79)	4168.05 (USD 654.38)	174.6 (USD 27.41)	630.33 (USD 98.96)	38.86 (USD 6.10)	339.48 (USD 53.30)	5859.55 (USD 903.72)
Moxifloxacin	667.94 (USD 104.87)	4257.63 (USD 668.45)	191.1 (USD 30.00)	1364.64 (USD 214.25)	10.47 (USD 1.64)	339.48 (USD 53.30)	6831.26 (USD 1053.59)
Levofloxacin	380.21 (USD 59.69)	4349.67 (USD 682.90)	128.37 (USD 20.15)	2582.84 (USD 405.51)	78.2 (USD 12.28)	339.48 (USD 53.30)	7858.78 (USD 1212.06)

## Data Availability

The original contributions presented in the study are included in the article/Appendix A, further inquiries can be directed to the corresponding authors.

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
