# Peer review of "Economic Evaluation of Nemonoxacin, Moxifloxacin and Levofloxacin in the Treatment of Early Community-Acquired Pneumonia with Possible Pulmonary Tuberculosis"

_ijerph, 2022, doi:10.3390/ijerph19084816_

Round 1

Reviewer 1 Report

The manuscript ijerph-1656661 is devoted to the actual problem of public health, namely the tconomic evaluation of quinolones in the treatment of early community-acquired pneumonia. The reviewed article is interesting for scholars and theme of the article meets the scope of the journal. Work is performed at sufficient scientific level and has good quality; the results of study are professionally interpreted. It can have important application value. The manuscript may be considered for publication after minor revision in International Journal of Environmental Research and Public Health. Prior publication of this manuscript following points needs to be addressed:

  • The figure 1 is of poor quality. It needs to be refined.
  • References list should be carefully checked and journal style policy should be strictly followed (citation rule for books, monographs and articles, all authors, etc).
  • It would be good to broaden conclusions in the context of a more detailed presentation of practical results and ways to resolve the problem.
  • Moderate English changes required. There are grammar/typing and orthographical errors in the manuscript.

My decision is minor revision

Author Response

Dear reviewer,

We sincerely thank you for your careful reading and examination, your constructive suggestions are of great help in improving the quality of our study (Economic evaluation of nemonoxacin, moxifloxacin and levofloxacin in the treatment of early community-acquired pneumonia with possible pulmonary tuberculosis).

We have carefully considered all professional comments from you and revised our manuscript accordingly. The manuscript has also been double-checked, and the typos and grammar errors we found have been corrected. In the following section, we summarize our responses to each comment from your reviewers. We believe that our responses have well addressed all concerns from the reviewers. We hope our revised manuscript can be accepted for publication.

Following are your advice and my modification:

Your advice:

  1. The figure 1 is of poor quality. It needs to be refined.
  2. References list should be carefully checked and journal style policy should be strictly followed (citation rule for books, monographs and articles, all authors, etc).
  3. It would be good to broaden conclusions in the context of a more detailed presentation of practical results and ways to resolve the problem.
  4. Moderate English changes required. There are grammar/typing and orthographical errors in the manuscript.

My modification:

  1. Thanks for your reminder. Based on your suggestion, we refined Figure 1, such as adjusting the size and specifications of the graphics, and we have replaced Figure 1 in our revised manuscript.
  2. We’re so sorry that due to our mistakes, there are many problems in the original references list. We have carefully checked the references list, and adjusted the format of authors, books, monographs in the references list according to the requirements of the journal based on your suggestions. Revised references list are provided in our updated manuscript.
  3. Thank you for your professional comment. Indeed, the conclusion part of the original article is too simple. Based on your suggestion, we have expanded the conclusions from both the research background and the problem addressed by the article. We updated the conclusion in the revised manuscript. For your convenience, I will briefly introduce the content of the updated conclusion here. As follows:  A.The summary of main two findings of our article, namely, nemonoxacin’s value in CAP and non-fluoroquinolone antibiotics’advantages compared to fluoroquinolone antibiotics. B Applications of two findings. That is, our findings may help health system make relevant policies and help clinicians in choosing optimal drugs in clinical medication. C The direction of future research. Such as using the society perspective to consider both direct and indirect cost of fluoroquinolone drugs in the treatment of CAP and misdiagnosed PTB.
  4. We are sorry for the inconvenience caused by us. We checked the full text again and fixed multiple grammar or typing errors. I believe there are no more similar problems in the revised manuscript.

Reviewer 2 Report

I thank the authors for the opportunity to review an article that presents a relevant topic.

The article presents an interesting structure and follows a logical sequence to show the main results. However, in my view, some gaps need to be resolved before the article can be considered.

1 - Literature has shown that indirect costs can significantly influence total costs. In this study, why were indirect costs not assessed? Justify.

2 - Present the cost driver.

3 - Why was a costing method already validated by literature not used, such as Time-driven activity-based costing - TDABC? 
See:

- Kaplan RS, Anderson SR. Time-driven activity-based costing. Harv Bus Rev. 2004;82(11):131–8 150.

 - da Veiga, C.R.P., da Veiga, C.P., Souza, A. et al. Cutaneous melanoma: cost of illness under Brazilian health system perspectives. BMC Health Serv Res 21, 284 (2021). https://doi.org/10.1186/s12913-021-06246-1

4 - What is the main motivation to compare the cost between the 03 drugs?

5 - The conclusion needs to be improved.

  • What are the main contributions of the study?
    - What are the theoretical, practical, and political implications of the study?
    - Also indicate suggestions for future work.

Author Response

Dear reviewer,

We sincerely thank you for your careful reading and examination, your constructive suggestions are of great help in improving the quality of our study (Economic evaluation of nemonoxacin, moxifloxacin and levofloxacin in the treatment of early community-acquired pneumonia with possible pulmonary tuberculosis).

We have carefully considered all professional comments from you and revised our manuscript accordingly. The manuscript has also been double-checked, and the typos and grammar errors we found have been corrected. In the following section, we summarize our responses to each comment from your reviewers. We believe that our responses have well addressed all concerns from the reviewers. We hope our revised manuscript can be accepted for publication.

Following are your advice and my modification:

Your advice:

1.Literature has shown that indirect costs can significantly influence total costs. In this study, why were indirect costs not assessed? Justify.

2.Present the cost driver.

3.Why was a costing method already validated by literature not used, such as Time-driven activity-based costing - TDABC?

4.What is the main motivation to compare the cost between the 03 drugs?

5 - The conclusion needs to be improved.

What are the main contributions of the study?

- What are the theoretical, practical, and political implications of the study?

- Also indicate suggestions for future work.

My modification:

  1. Thanks for yourprofessional question. We have highlighted this in the Discussion section of the article (lines 342-346). And the main reasons for not considering indirect cost are as follows: A. One of the main purposes of research is to help health systems in policy decisions, so we chose the perspective of Chinese health system. And once this perspective is selected, only direct medical costs associated with the health system can be considered, indirect costs such as transportation and lost work cannot be consider as they have little connection to the health system, so for this reason we have not considered indirect cost. B. Due to the short time period of this study (194 days), and the treatment time for pulmonary tuberculosis reached 180 days (there was no difference in the treatment regimens for tuberculosis among 3 groups), which means that there is little difference between indirect cost across groups, so ignoring the indirect costs would not affect the conclusions of our study. C. But in order to calculate cost more precisely, indirect cost is needed to be considered, though it won’t affect our conclusion. Thus, in our future research, we plan to conduct further cost research and use a societal or individual patient perspective to incorporate both direct and indirect costs for economic evaluation based on your reminder.
  2. Thanks for your reminder. We have added cost driver according to your advice, and please see the Table 3 in our updated manuscript for more details about cost drive.
  3. Thanks for your professional and meaningful question. Time-driven activity-based costing is a scientific method, and its application in the health economics is becoming more and more extensive nowadays. In my view, it can be used in costing under a hospital perspective. But in our study, we measure costs from Chinese health system perspective. Under this perspective, costs represent the resource consumption of the health system through market exchanges, so there is no need to dismantle it to the accounting level according to Chinese Guidelines for Pharmacoeconomics Evaluation[1]. The same costing approach was taken in similar cost-effectiveness analysis studies[2-5], so we believe our cost estimates are reasonable under our chosen perspective.

[1]Guoen L. Chinese Guidelines for Pharmacoeconomics Evaluation 2020. China Market Press 2020.

[2]Du X, Han Y, Jian Y, et al. Clinical Benefits and Cost-Effectiveness of Moxifloxacin as Initial Treatment for Community-Acquired Pneumonia: A Meta-Analysis and Economic Evaluation. Clin Ther. 2021 Nov;43(11):1894-1909.e1.

[3]Egger ME, Myers JA, Arnold FW, et al. Cost effectiveness of adherence to IDSA/ATS guidelines in elderly pa-tients hospitalized for Community-Aquired Pneumonia. BMC Med Inform Decis Mak. 2016 Mar 15;16:34.

[4]Lloyd A, Holman A, Evers T. A cost-minimisation analysis comparing moxifloxacin with levofloxacin plus ceftriaxone for the treatment of patients with community-acquired pneumonia in Germany: results from the MOTIV trial. Curr Med Res Opin. 2008 May;24(5):1279-84.

[5]van Werkhoven CH, Postma DF, Mangen MJ, et al. Cost-effectiveness of antibiotic treatment strategies for community-acquired pneumonia: results from a cluster randomized cross-over trial. BMC Infect Dis. 2017 Jan 10;17(1):52.

  1. Thanks for your question under a responsible attitude. The motivation to compare the cost between the three drugs is as follows, and we highlighted this point in the Discussion (lines 397-412) and Conclusion section (lines 348-361) in the revised manuscript.   A . Nemonoxacin is a non-fluoroquinoloneantibiotic, while moxifloxacin and levofloxacin are fluoroquinolone antibiotics. We wanted to determine the economic advantages of non-fluoroquinolone over fluoroquinolone antibiotics in the differential diagnosis of pulmonary tuberculosis by comparing their costs.  B. From the perspective of Chinese health system, we want to compare the cost-effectiveness between the three drugs in order to help with policy development and clinical medication. While the utilities of three intervention were almost the same, so the cost-saving intervention tend to be more economical. And this is another reason for us to compare costs of the three drugs.
  2. We are sorry for leaving out related expression in our article which caused addition work for you. Based on your suggestions, we have rewritten the conclusion (lines 348-361) in order to cover the points you mentioned. For your convenience, we provided answers for your questions as follows:  A. From the perspective of Chinese health system, we confirmed that nemonoxacin is cost-effectiveness compared with moxifloxacin and levofloxacin, and we revealed the values in money and utility of non-fluoroquinolone antibiotics in the differential diagnosis of pulmonary tuberculosis compared with fluoroquinolone antibiotics. These findings can help health system make relevant policies in medicare negotiations and help clinicians in clinical medication. B. As you pointed out, indirect costs were not considered in our study due to the chosen research perspective. Although ignoring the direct cost would not affect the research conclusion, in order to calculate the cost more accurately, for further research, an economic evaluation from the perspective of the society or patients using professional costing method (such as TDABC you mentioned) are needed so that both direct and indirect cost can be covered. And future study are advised to consider more non-fluoroquinolone and fluoroquinolone antibiotics at the same time.

Reviewer 3 Report

Dear Authors,

It is very interesting topic and valuable.

  1. This part should be extended: (lines 78-80) "In total, doctors from 31 authoritative hospitals were questioned, which were located in Beijing, Shanghai, Hainan, Gansu, Wuhan, and other cities in the east, middle, and west of China. The clinical diagnosis and treatment period covered was 20±8 years".         .                                                          It is difficult to figure out the representativeness level of this research. It would be valuable to provide information how many hospitals are in total in China. How they have been chosen? What criteria determined that these hospitals were chosen ? And how many cases (patients) were analyzied? 
  2. Why you have not taken indirect cost and can they differ according to the treatment option? 
  3. What is the application of your research? It is not underlined in your article. 
  4. What is the direction of future research ? It is not mentioned in your article
  5. The range of references is a little bit limited.

Author Response

Dear reviewer,

We sincerely thank you for your careful reading and examination, your constructive suggestions are of great help in improving the quality of our study (Economic evaluation of nemonoxacin, moxifloxacin and levofloxacin in the treatment of early community-acquired pneumonia with possible pulmonary tuberculosis).

We have carefully considered all professional comments from you and revised our manuscript accordingly. The manuscript has also been double-checked, and the typos and grammar errors we found have been corrected. In the following section, we summarize our responses to each comment from your reviewers. We believe that our responses have well addressed all concerns from the reviewers. We hope our revised manuscript can be accepted for publication.

Following are your advice and my modification:

Your advice:

1.This part should be extended: (lines 78-80) "In total, doctors from 31 authoritative hospitals were questioned, which were located in Beijing, Shanghai, Hainan, Gansu, Wuhan, and other cities in the east, middle, and west of China. The clinical diagnosis and treatment period covered was 20±8 years". It is difficult to figure out the representativeness level of this research. It would be valuable to provide information how many hospitals are in total in China. How they have been chosen? What criteria determined that these hospitals were chosen? And how many cases (patients) were analyzied?

2.Why you have not taken indirect cost and can they differ according to the treatment option?

3.What is the application of your research? It is not underlined in your article.

4.What is the direction of future research ? It is not mentioned in your article

5.The range of references is a little bit limited.

My modification:

1.Thanks a lot for your suggestions with a serious and responsible attitude to our manuscript. The part (lines 78-80) should indeed be extended. First, we are very regretted making a mistake here, that is, “31 authoritative hospitals” ought to be replaced by 31 doctors. According to your advice, we have refined and extended expression in our updated manuscript (lines 82-89). For your convenience, the answers for your questions and main changes of this part are concluded as follows:

Our criteria for selecting doctors and hospitals is to make sure our data is reliable and can represent the overall level of China. And we conducted this survey by distributing electronic questionnaires. Thus, the doctors we chose are all experienced in clinical diagnosis and treatment. In addition, we chose representative hospitals from all around China, that is, authoritative hospitals in the eastern, middle and western regions of China. Finally, 31 doctors from nationwide authoritative hospitals were questioned, which were located in Beijing, Shanghai, Wuhan, and other cities. The mean years of the clinical treatment of these doctors was 20±8 years, and the average age was 45±8 years old.

2.Thanks for your professional question. We have highlighted this in the Discussion section of the article (lines 342-346). And the main reasons for not considering indirect cost are as follows:

A.One of the main purposes of research is to help health systems make relevant decisions, so we chose the perspective of Chinese health system. And once this perspective is selected, only direct medical costs associated with the health system can be considered, indirect costs such as transportation and lost work cannot be consider as they have little connection to the health system, so for this reason we have not considered indirect costs.

B.Due to the short time period of this study (194 days), and the treatment time for pulmonary tuberculosis reached 180 days (there was no difference in the treatment regimens for tuberculosis among 3 groups), which means that there is little difference between indirect cost across groups, so ignoring the indirect costs would not affect the conclusions of our study.

C.But in order to calculate cost more precisely, indirect cost is needed to be considered, though it won’t affect our conclusion. Thus, in our future research, we plan to conduct further cost research and use the society or patients perspective to incorporate both direct and indirect costs for economic evaluation.

3.We are sorry that we left out relevant expressions in the article and caused you extra workload. Based on your advice, we have added the application of the research in the Abstract (lines 33-37) and Conclusion (lines 357-359) sections of the revised manuscript. The application of your research is mainly as follows:

From the perspective of Chinese health system, we confirmed that nemonoxacin is cost-effectiveness compared with moxifloxacin and levofloxacin, and we revealed the values in money and utility of non-fluoroquinolone antibiotics in the differential diagnosis of pulmonary tuberculosis compared with fluoroquinolone antibiotics. These findings can help health system make relevant policies in medicare negotiations and help clinicians in choosing optimal drugs in clinical medication.

4.Indeed, we are lack of the direction of future research. We have added related expression in the Conclusion section (lines 348-361) of the revised manuscript. For your convenience, I state the direction of future research as follows:

      As you pointed out, indirect cost were not considered in our study due to the chosen research perspective. Although ignoring the indirect cost will not affect the research conclusion, in order to calculate the cost more accurately, we will conduct further research, try to make an economic evaluation from the perspective of the society or patients so that both direct and indirect cost can be covered, and consider more non-fluoroquinolone and fluoroquinolone antibiotics at the same time.

5.Thanks for your reminder, the original range of references is limited. According to your advice, we revised our article, added some references after some expressions(lines 55,186) in the article and removed some useless references. Finally, we added 4 references compared to the previous ones.

Round 2

Reviewer 2 Report

Dear authors, 

Thank you for the effort to respond point by point to my comments. The article now has the potential to be published. Congratulations.

Reviewer 3 Report

All comments have been taken into account. The quality of paper improved.